# Testing the Minimum System Entropy and the Quantum of Entropy

**DOI:** 10.3390/e25111511

**Published:** 2023-11-03

**Authors:** Uwe Hohm, Christoph Schiller

**Affiliations:** 1Institut für Physikalische und Theoretische Chemie, Technische Universität Braunschweig, Gaußstr. 17, 38106 Braunschweig, Germany; 2Motion Mountain Research, 81827 München, Germany

**Keywords:** minimum system entropy, quantum of entropy, lower entropy limit, third law of thermodynamics, entropy quantization, black hole, Boltzmann constant

## Abstract

Experimental and theoretical results about entropy limits for macroscopic and single-particle systems are reviewed. All experiments confirm the minimum system entropy S⩾kln2. We clarify in which cases it is possible to speak about a *minimum system entropy*
kln2 and in which cases about a *quantum of entropy*. Conceptual tensions with the third law of thermodynamics, with the additivity of entropy, with statistical calculations, and with entropy production are resolved. Black hole entropy is surveyed. Claims for smaller system entropy values are shown to contradict the requirement of observability, which, as possibly argued for the first time here, also implies the minimum system entropy kln2. The uncertainty relations involving the Boltzmann constant and the possibility of deriving thermodynamics from the existence of minimum system entropy enable one to speak about a general *principle* that is valid across nature.

## 1. Introduction

In thermodynamics, the concepts of *minimum system entropy* and of *quantum of entropy* are rarely mentioned. Only a small number of authors have suggested that the Boltzmann constant *k* plays the role of a quantum of entropy that contains and implies all of thermodynamics. Notable examples include Zimmermann [1,2,3,4,5] and Cohen-Tannoudji [6].

In this article, we argue that the Boltzmann constant $k\approx 1.4\cdot 10^{-23}\;J/K$ introduced by Planck is not merely a conversion factor relating energy and temperature, but that it has a deeper meaning in nature: *k* fixes, with a prefactor ln2, the *lower limit* to system entropy. Conversely, the lower limit kln2 for system entropy can be seen as *characterizing* thermodynamics. We show this by exploring two questions.

First, is there a ‘quantum of entropy’ in nature at all? We explore which systems have a lower limit for entropy, even though most systems do not have quantized entropy values.

Secondly, is thermodynamics characterized by the Boltzmann constant *k* in the same way that special relativity is characterized by the speed of light *c* and quantum theory is characterized by the quantum of action *ℏ*? In other words, we test whether there is a *principle* of limit entropy kln2.

In the past, different authors have arrived at different conclusions. We review the published arguments in favor and against, give an overview of the results from low-temperature physics up to quantum gravity, and conclude with a structured and coherent summary of the quantum of entropy and its domain of application. In Section 8, system entropy is even shown to be bounded *a priori*, using an argument that appears to be new. In the discussion, we find it often useful to be somewhat imprecise and to call both *k* and kln2 the ‘quantum of entropy’.

The expression kln2 derives from a thermodynamic system with two microstates. Can this be the smallest system entropy? Can there be a quantum of entropy in nature at all? We start with two issues that appear to provide a negative answer.

## 2. The Definition of Entropy

The physical observable called *entropy* measures disorder. A useful description is to describe entropy as the carrier of thermal energy, in the same way that a fuel is a carrier of chemical energy or momentum is a carrier of kinetic energy [7]. Like all energy carriers, entropy is also an extensive quantity. As is well known, in closed systems, the value of entropy never decreases. This second law of thermodynamics describes and allows dissipative processes, i.e., processes which *produce* entropy, and forbids processes that *destroy* entropy.

In classical thermodynamics, entropy is a continuous quantity, without any minimum value. Only statistical mechanics changes the situation.

Furthermore, in classical thermodynamics, entropy is regularly stated to be defined only up to an additive constant [8,9,10,11,12]. However, this statement has been questioned by Steane [13]. In any case, in statistical physics, Boltzmann defined entropy as a consequence of the number of microstates that lead to the same macrostate. In statistical physics, there is no freedom of choosing the additive constant in the definition of entropy.

An overview of different definitions of entropy is given by Šafránek et al. [14]. They summarize the definitions in the unifying concept of observational entropy. Entropy remains a fascinating topic of research to this day. Its applications range from self-organization [15] to the configuration of optical networks [16,17].

*In short*, in the following, the term ‘entropy’ refers to the observational entropy of a closed system at equilibrium, except when mentioned otherwise. The second law by itself does not provide a lower entropy limit, but does not exclude it either. However, a further argument seems to contradict a minimum entropy value kln2 in nature.

## 3. The Third Law of Thermodynamics

Starting in 1905, Nernst deduced a theorem about entropy that today is called the third law of thermodynamics [18]. He formulated it in various equivalent ways [19]. Two of Nernst’s formulations are:

The *entropy change* associated with a chemical or physical transition between condensed phases approaches *zero* when the temperature approaches absolute zero.

Absolute zero temperature cannot be reached in a finite number of steps.

In the years following its discovery, the third law was rephrased in additional ways. A frequently used one is attributed to Einstein [19]:

The *entropy* of a system approaches a *constant value* when its temperature approaches absolute zero.

Another formulation proposed by Planck is popular. He presented it in reference [20], where he wrote: “Entropie eines jeden chemisch homogenen (§67), dauernd ungehemmt im inneren Gleichgewicht befindlichen Körpers von endlicher Dichte nähert sich bei bis zum absoluten Nullpunkt abnehmender Temperatur einem bestimmten, vom Druck, vom Aggregatzustand usw. sowie von der speziellen chemischen Modifikation unabhängigen Wert. Da die Entropie bisher nur bis auf eine willkürliche additive Konstante definiert ist, können wir unbeschadet der Allgemeinheit diesen Grenzwert gleich Null setzen”. *Translation by the authors:* The entropy of any chemically homogeneous (§67), body of finite density and in permanent uninhibited inner equilibrium approaches, when the temperature decreases to absolute zero, a specific value that is independent of pressure, physical state, etc., and of the particular chemical modification. Since the entropy was defined so far only up to an arbitrary additive constant is defined, we can set this limit, without prejudice to generality, equal to zero. Planck also wrote in reference [21] “Dieselbe besagt, daß die Entropie eines kondensierten (d.h. festen oder flüssigen) chemisch einheitlichen Stoffes beim Nullpunkt der absoluten Temperatur den Wert Null besitzt […]”. *Translation by the authors:* It states that the entropy of a condensed (i.e., solid or liquid) chemically homogeneous substance has the value zero at the zero point of the absolute temperature […]. Planck’s result is usually stated as:

At zero temperature, the *entropy* of a chemically homogeneous body in equilibrium is *zero*.

In this last formulation, the details are important. For example, in the case of glasses, which have a high configuration entropy, equilibrium is not attained, and the third law can only be formulated with the unattainability of zero temperature [22]. As another example, a crystalline solid is stated to have zero entropy only if it is perfect, without any impurities, dislocations, or any other crystal defects, and with all nuclear spins locked against each other.

In addition, all formulations of the third law are valid in the *thermodynamic limit*, i.e., for systems that have an infinite number of particles, infinite volume, but constant density. Presentations, summaries and research on the third law of thermodynamics, its various formulations and their differences are found, for example, in references [23,24,25,26,27,28,29,30,31,32,33,34].

*In short*, before using the third law to argue against a minimum system entropy, the *conditions* and the *precision* for the predicted *zero* entropy value appearing in the various formulations of the third law must be checked. Given that the third law was derived using the thermodynamic limit and classical physics, several questions arise.

–Is the third law confirmed by all experiments?–Is the third law valid in quantum theory?–Is the third law valid for small systems, and in particular, for single particles?

The exploration will show that for each question there are systems that do *not* follow the *naive* third law stating that system entropy vanishes at vanishing temperature. Systems never actually have vanishing entropy and there indeed is a ‘quantum of entropy’ in nature. A first hint arose already long ago.

## 4. A Smallest Entropy Value?

Szilard was the first researcher to suggest, in 1929 [35], that the smallest system entropy occurs in single-particle systems and plays a role in nature. Using simple thought experiments while exploring the details of Maxwell’s demon, he deduced the value kln2 for the entropy *change* in the case that a free particle is forced to choose between two possible enclosed volumes of the same size. The numerical factor ln2≈0.69314… is due to Boltzmann’s expression for entropy S=klnΩ in a situation where the particle chooses between two equal volumes. In modern language, the factor ln2 expresses that the entropy in a system with two microstates is described by a single bit of information.

Szilard thus explained that there is a quantized entropy *change*, i.e., a finite entropy *step* in nature, whenever a *single* particle changes from a situation with one possible state to a situation with two possible states, yielding an entropy change of kln2. Exploring the measurement process performed by Maxwell’s demon, he writes “größer dürfte die bei der Messung entstehende Entropiemenge freilich immer sein, nicht aber kleiner”, or “the amount of entropy arising in the measurement may, of course, always be greater [than this fundamental amount], but not smaller”. His description needs several clarifications.

–Szilard does not discuss entropy values *per particle* in multi-particle systems. He leaves open whether a smallest or largest entropy value or entropy change per particle exists in nature.–Szilard does not discuss the entropy of *macroscopic systems.* He leaves open whether a *smallest* system entropy *value* exists in nature. Szilard also leaves open whether a smallest value for the entropy *change* for a macroscopic system exists in nature.–Szilard discusses the case of a *one-particle system* with a small number of microstates. He suggests a *characteristic* value for the entropy *change* for small numbers of microstates. He does show—when discussing the first equation in his paper—that a smaller value of entropy does not arise. In contrast, high numbers of microstates allow both smaller and larger values of entropy change.

Generally speaking, Szilard highlights the relation between entropy steps and the quantization of matter. Without particles, entropy steps would not occur, and the Boltzmann constant *k* would not arise.

Not long ago, Szilard’s thought experiment was realized in the laboratory by Koski et al. [36]. Also, the quantum thermodynamics experiments based on quantum dots by Durrani et al. [37], Abualnaja et al. [38] and those based on nuclear magnetic resonance by Vieira et al. [39] confirm Szilard’s results, including the value kln2.

*In short*, the paper by Szilard does *not* make *clear* statements on the importance or existence of a *quantum of entropy* or a *minimum entropy* kln2. The following sections explore whether and under which conditions Szilard’s value—or a similar value such as *k* itself—is a useful concept for describing nature. This exploration can be divided into three cases. First, any proposed quantum of entropy must be compared to the observed values of (1) the entropy and the entropy change *per particle* in macroscopic systems. Then, a quantum of entropy must be compared to the observed values of (2) the *total* entropy and the *total* entropy change for *large systems*. Finally, a quantum of entropy must be compared to the observed values of (3) the entropy and the entropy change for a *single particle*. This step-by-step approach avoids making too general statements about a quantum of entropy too quickly [40]. For each case, experiments and calculations provide insights.

## 5. The Entropy per Particle in Macroscopic Systems

The concept of entropy *per particle* has two possible meanings. The first, assumed here, is the total entropy of a system divided by the number of particles. The second meaning is the change in total system entropy when one particle is added. This second meaning is not explored in this study.

In experiments with *macroscopic matter* systems at low temperatures, entropy values *per particle* much lower than kln2≈0.69k have been measured. For instance, while lead has an entropy per atom of 7.79k at room temperature, diamond has an entropy per atom of 0.29k, which is lower than the proposed lower limit for entropy. At a temperature of 1 K, solid silver has an entropy per atom of 8.5·$10^{-5}\;k$ [41,42].

Similarly, in Bose–Einstein condensates, entropy values per atom have been measured to be as low as 0.001k [43], with a total entropy of about 1000k per million particles. It is planned to achieve even lower values for the entropy per particle in future microgravity experiments [44]. Fermion condensates show similarly small values for the entropy per atom [45,46]. Also, superfluid helium-II can be cited as an example of a system with an almost negligible entropy per particle [47,48]. As another example, ${}^{3}\mathrm{He}$ has, in the region between 0.01 K and 1 K, an entropy of at least kln2 per atom, due to the nuclear spins; however, at much lower temperatures, when the material solidifies and the spins interact, the entropy is much lower [49].

Calculations of entropy in specific atomic systems confirm the experiments just cited. They show that in solids, in contrast to gases, the entropy per particle can indeed be *much* lower than *k*. The calculations are involved but confirm the observations. Ludloff [50,51,52] used quantum statistics to deduce S(T=0)=0 for macroscopic bodies. Dandoloff and Zeyher [53] determined that the entropy per particle approaches zero as T→0. A similar result was obtained by De Leo et al. [54].

Also, the entropy of *photon ensembles* has been explored in detail. Given the observation that light carries entropy, any experiment with light that shows photon behavior can be used to deduce that individual photons carry entropy. However, the value of the quantum of photon entropy needs to be determined.

The entropy of the *black body photon gas* has been presented by many researchers, for example in the references [2,26,55,56]. For example, Zimmermann explained that the entropy *S* of a black body photon gas with *N* photons is strictly larger than kN because the entropy is not only due to the (average) number of photons *N* but also due to their momentum distribution. The result for the thermal photon gas is
(1)$$S=\frac{2\pi^{4}}{45\;\zeta (3)}kN\approx 3.602\;kN\;\;,$$ζ(3) being the Riemann zeta function. Thus, in a black body photon gas, like in a usual matter gas, the entropy per particle is larger than *k*.

The situation changes in *beams of light*, where all photons have the same frequency and similar directions [57]. Scully [58] estimated the entropy change of a laser with one *additional* photon at the threshold and found that the change can easily be as low as $10^{-6}\;k$. However, as pointed out by Li et al. [59], the definition of single-photon entropy is involved and not unique. At equilibrium, the entropy *s* of a single monochromatic photon with frequency ω can be argued to be s=ℏω/T. According to this relation, for visible light at room temperature, one gets s≈100 k. At the same time, the full entropy *S* of such a light field can be calculated to be of the order of S=(1+N)ln(1+N)k−NlnNk≈ln(N+1)k. As a result, the entropy *s* per photon in a monochromatic light beam with large *N* is s≈klnN/N, which is again much smaller than *k*.

*In short*, this section covered case (1) given at the end of the previous Section 4: in macroscopic, multi-particle systems—i.e., in the thermodynamic limit—both for matter and radiation, experiments show that there is *no* smallest value for the *entropy per particle*, and *no* smallest value for the *entropy change per particle.* This result is as expected from thermodynamics: when the particle number increases, the entropy steps, or entropy changes, decrease without any positive lower limit. In particular, they can be several orders of magnitude smaller than kln2. Therefore, we now turn to *total* system entropy.

## 6. Quantum Theory and the Third Law

The case (2) listed at the end of Section 4 is the exploration of possible lower limits for *system entropy* and for the *change* of system entropy for *macroscopic* systems. We begin by reviewing the results of quantum theory about the third law of thermodynamics.

Already, Einstein noted the necessity of considering *quantum theory* to prove the third law of thermodynamics [60,61]. Indeed, quantum theory appears to confirm that the entropy of a condensed matter system vanishes at zero temperature, provided that its ground state is unique, and thus not degenerate [50,51,52]. Moreover, Wehrl, in his influential review, stated that the entropy of a pure quantum state is exactly zero [62]. These authors concluded that S(T=0)=0 for degeneracy g=1. Dandoloff and Zeyher argue that a perfect crystal in its ground state has only a single microstate and therefore has vanishing entropy [53]. The same point was made using the modern approach of quantum thermodynamics [63]. All these results were deduced in the thermodynamic limit.

The validity range of the third law has been explored by several authors. An example is the discussion by Lawson [64]. He found no experimental deviation from the third law. Pañoz and Pérez [65] compared the experimental results on entropy *S* to the Sackur-Tetrode equation. Within the measurement uncertainties, very good agreement was observed, provided that S(T=0)=0 is chosen. The study by Loukhovitski et al. [66] confirmed that solid nanoparticles follow the third law.

Statistical calculations yield similar results. Scully calculated S(T=0)=0 for a Bose–Einstein condensate [58]. Ben-Naim claimed that S(T=0)=0 in references [67,68]. The mathematical analysis by Belgiorno confirms the Planck version of the 3rd law, $\lim_{T\to 0^{+}}S\left(T\right)=0$ [69,70]. According to Shastry et al. the third law is also valid in open quantum systems [71]. Steane presented an alternative route to obtaining S(T=0), without the third law or quantum mechanics [13]. He also stated that S(T=0)=0 is observed in many cases.

In contrast to the mentioned authors, some theorists argue that the third law with its expression S(T=0)=0 is a convention. This was stated by Klotz [31] and by Falk [72]. Griffiths went further and claimed that the entropy does not vanish even for ground state non-degeneracy [73] because the microstates near the ground state also play a role in the calculation of entropy. Aizenman and Lieb disagreed and argued that the validity of the third law is decided completely in terms of ground-state degeneracies alone [74]. However, they also stated that their argument is not completely tight.

*In short*, experiments and many calculations *for the thermodynamic limit* confirm the third law of thermodynamics, with its zero entropy and zero entropy change (within the measurement limits) at zero temperature. In contrast, quantum theory does *not* confirm the third law for systems that do *not* realize the thermodynamic limit, and in particular, for systems made of a single particle.

## 7. Entropy and Entropy Change in Single-Particle Systems

The central statement of this article is that minimum system entropy kln2 exists in single-particle systems. Such a claim has to be tested in experiments, for the cases of radiation, of matter, and of information.

Entropy and information are related. The physics of information was studied already by Brillouin. In his influential book [75], he explored the idea that the photon is a quantum of information that carries an entropy *k*. In 1983, Pendry showed that information flow is entropy flow divided by kln2 [76], a relationship that was further clarified by Blencowe and Vitelli [77]. Also, according to Ben-Naim, the Shannon measure of information is connected to the entropy by applying a factor of kln2 [67].

Above all, Pendry showed that the conductivity of especially narrow channels is quantized. This has led to a large amount of experimental work. Many experiments during the past decades detected—in analogy to quantized electric conductance in multiples of $2e^{2}/h$— *quantized thermal conductance* in multiples of $\pi^{2}k^{2}T/3h$ (*T* being the temperature) and *quantized entropy conductance* in multiples of $\pi^{2}k\nu /3$ (ν being the carrier frequency).

Meschke et al. observed the quantization of heat conductance via *photons* [78]. This implies that in their experiment entropy transport occurs via a sequence of single photons. In their experiments, they confirmed the quantized conductivity deduced by Pendry. In this way, Meschke et al. confirmed that in quantized entropy transport with photons, each photon carries an entropy of the order *k*. The result confirms the results cited earlier showing that single photons carry entropy. Photon entropy has also been studied in the context of laser radiation, photosynthesis, and the laser cooling of matter. Kirwan [79], Van Enk and Nienhuis [80] (“The [produced] entropy per photon is therefore […] larger than k”), and Chen et al. [81,82,83] argued in detail that a single photon, one that is *not* part of a photon field, always carries a quantum of entropy of the order of *k*.

A selection of experimental observations of quantized entropy flow using *phonons* can be found in references [84,85,86,87,88,89,90]. Quantized flow was also observed for *electrons* [91,92] and *anyons* [93]. The numerical value of the quantum limit of heat flow via electrons was confirmed experimentally in 2013 [87]. All these experiments confirmed that for a single quantum channel, quantum effects provide a lower limit for entropy flow.

Also, in two-dimensional electron gases, quantized entropy per particle was predicted and then observed [94,95].

Calculations of the quantized entropy conductance were discussed in detail by Márkus and Gámbar [96] and Strunk [97]. Their analyses confirmed that for single channels, entropy transport is quantized. In contrast, conductance is not quantized for ‘wide’ channels or for channels with non-ballistic transport. However, these other cases do not invalidate the general argument.

Theoretical single-particle thermodynamics does not appear to have explored the topic of the smallest entropy of single particles. For example, Bender et al. deduced that a heat engine for a single particle can be realized [98,99]. However, they deduced no statement about the existence of a quantized entropy change, neither in favor of nor against it [100,101]. Also, the experimental realization [102] of a single-atom heat engine makes no such statement. Some treatments of single-particle thermodynamics even explicitly disagree with a smallest system entropy, such as Ali et al. [63], who stated that the entropy of a single particle vanishes at zero temperature when it is coupled to a bath. We resolve this issue in the next section.

*In short*, experiments confirm that, in closed single-particle systems, a smallest entropy value exists and is observable. This is in full contrast to the case of the thermodynamic limit. In the case of single photons, single phonons, and single electrons, a quantum of entropy is observed:

▷ Single particles carry a finite entropy that is never lower than kln2.

This is an important experimental finding. *A system entropy limit exists because radiation and matter are made of particles*. No closed system with a total entropy smaller than kln2 has been observed. We note that the result places no limit on entropy *changes* or *steps*; these can be arbitrarily large or infinitesimally small. This summary settles the case (3) given at the end of Section 4, which asked about the entropy of single particles. Nevertheless, the concept of minimum system entropy must be checked in more detail.

## 8. Minimum System Entropy and Observability

Summarizing, *no* experiment has ever found a deviation from the third law for macroscopic systems, and at the same time, *no* experiment has ever confirmed the third law for single-particle systems. In other terms, in the thermodynamic limit, entropy and entropy change are *effectively continuous* quantities. In contrast, for *single particles*, all experiments—such as the experimental tests of Szilard’s experiment and of information erasure listed above—and most calculations agree on a lower entropy limit. But not all.

One general argument is regularly provided against a non-vanishing lower limit for the entropy of macroscopic systems. It is regularly stated that a quantum system in a *non-degenerate* ground state does have vanishing total entropy: for such a ground state, the expression S=klnΩ, where Ω is the number of microstates, implies that Ω=1 and thus that the zero-point entropy vanishes exactly. It turns out that there are at least two fundamental reasons why this popular argument is incorrect.

First, given the measurement uncertainties in the measurement of any (quasi-) continuous quantity, one *cannot* prove that the quantity has a zero value. This is especially the case for a quantity such as entropy which is, by definition, always positive. Quantum theory always yields non-zero measurement uncertainties, also for entropy and temperature. These measurement uncertainties are related to Boltzmann’s constant *k*, as shown later on. Measurement uncertainties imply that an *exactly vanishing* entropy value cannot be confirmed in any experiment. (This is in contrast to positive invariants, such as the speed of light, which can be confirmed within measurement errors.) Even at the lowest temperatures, measurement errors allow only to deduce an upper limit for the entropy. It is thus necessary to check the issue in actual experiments.

As all the experimental papers cited in this text show, for a closed system at equilibrium, no system entropy smaller than kln2 has ever been observed. This result is important because many experiments have measurement uncertainties that are *smaller* than the minimum entropy value. As shown below, the impossibility of measuring smaller entropy values for closed systems is expected also from the thermodynamic uncertainty relations.

A second reason speaks against the existence of a system with only one microstate, and thus with vanishing entropy. Every physical system has a basic property: it is *observable.* Any observation is an interaction. For example, observing a car implies scattering photons from it. As another example, observing a mass can mean placing it on a scale. Every observation and every measurement requires an interaction with the measurement apparatus. The interaction implies that the system being observed can have *several* microstates. (The existence of interaction-free measurements does not change the argument.) In particular, the basic property of observability implies that every physical system can have at least *two* states: it is being observed and it is not. These two states are different microstates that correspond to the same macrostate. In fact, the case of two microstates must be seen as the bare minimum. In practice, a system can often be observed by different observers, so that it can have even more ground microstates. In other words, the case with state multiplicity Ω=1 is *impossible* for an *observable* system.

▷ Observability implies a smallest entropy value of kln2 for every system.

A striking way to put this result is the following: *only an unobservable system can have zero entropy*. The argument just given resembles a well-known statement by DeWitt [103]. He stated that every system is either there or not, and that, therefore, any system must have at least an entropy kln2. To the best of our knowledge, the argument based on observability is not found in the literature.

*Only a system that is never observed and never interacts with the environment could have vanishing entropy*. However, no such system exists, because these conditions contradict the concept of ‘system’. The conditions even contradict the concept of objectivity: Unobservable or non-interacting systems are not part of the natural sciences. (Again, the existence of interaction-free measurements does not change the argument.) To complete the argument, it should be noted that the ‘universe’ itself is not a system in the sense used here.

Physical systems have a minimum entropy given by kln2. However, the number of publications mentioning the minimum system entropy value in nature is surprisingly small. Natori and Sano [104], Ladyman et al. [105] or Norton [106], who explore the entropy of computation, prefer to state that the limit applies to entropy change. However, as mentioned above, this result is questionable, particularly when the number of microstates Ω is large and changes by only a small value. In other words, observability does not seem to allow deducing the smallest value for entropy *change*. This impossibility is also an expected consequence of the third law of thermodynamics.

*Minimum system entropy does not depend on the definition of entropy*. Experimentally, entropy is a uniquely defined concept. In theoretical physics one can explore Boltzmann entropy, Shannon entropy, von Neumann entropy, Tsallis entropy [107,108], and Renyi entropy [109]. While most of these types of entropy seem to be bounded by kln2 for systems of one particle, the present work makes this statement first of all for experimentally observed entropy values [14].

There still remain many ways to measure entropy values *smaller* than kln2. In particular, *open* systems and *entangled* systems can lead to much smaller entropy values. The minimum system entropy is only valid for *closed* systems at equilibrium.

*In short*, experiments and fundamental arguments confirm

▷ System entropy is limited by S⩾kln2.

The result is valid generally, for single-particle and for macroscopic thermodynamic systems. (Closure and equilibrium are assumed.) In particular, the minimum system entropy follows from the possibility of *observing* any physical system. The limit on system entropy is valid by definition and is independent of the substance and of the number and type of degrees of freedom of the system. It seems that this argument is not found in the literature so far. Despite this result, several arguments against the minimum system entropy must be discussed.

## 9. Two-Level-Systems and Entropy Calculations for Similar Idealized Systems

The minimum system entropy limit is rarely mentioned in books and publications. The reason is that the minimum system entropy contradicts many simple arguments and calculations. The simplest counter-argument, mentioned above, is the mentioned prejudice that systems have a single ground state. But there are more such arguments.

Any *two-level system* at finite temperature, with a small energy difference between the two levels, yields an entropy value smaller than kln2. But if the two-level system really consists only of a single particle, the interaction with the thermodynamic system that produces the two levels *cannot* be neglected. If one does neglect it, small or even vanishing entropy becomes possible, leading to calculation results that are in contrast to the real world.

Many experiments have been performed for matter systems with only two possible states. This topic became popular in 1961 when Landauer stated that the entropy required to erase one bit of information is at least kln2 [110]. Extracting entropy or erasing information from a macroscopic system requires energy. For a *memory with one bit*, and thus two states at the same energy, the required energy is defined by the entropy kln2 and the temperature of the system. Numerous experiments with glass beads in a double-well potential and with many other systems have confirmed this entropy value within the experimental uncertainties. The entropy value has also been confirmed for optical and for magnetic storage systems. An overview is provided in the book [111]; specific experiments are presented in references [112,113,114].

Do the astonishing experiments confirming Landauer’s ideas also confirm the minimum system entropy? *No, they do not*. The experiments *do* confirm that a system with two microstates has an entropy kln2. But they do *not* confirm minimum system entropy. In fact, Landauer even states, for the case that the initial state corresponds to a logical “1” or a logical “0”: “The well-defined initial state corresponds, by the usual statistical mechanical definition of entropy, S=klogW, to zero entropy”. Landauer thus states that a zero entropy state is possible when a system is in a well-defined and non-degenerate ground state. However, this statement is an idealization.

Any actual two-level system used in computing, whether in a transistor, in a magnetic film, or in any other memory, needs matter to generate the two energy levels. Therefore, even if the bit in memory is realized by a single particle, that particle is *not* a closed system, but an open one, even if the particle is not disturbed. And the minimum system entropy statement does not apply to *open* systems. Indeed, the system forming the memory does not have zero entropy.

Another example that shows how idealizations can be misleading is the case of a single quantum particle in a box. A single atom in a box has a single, non-degenerate ground state. Such an atom is regularly seen as a system with a single microstate, therefore with zero entropy, and thus apparently violating the minimum system entropy. But again, the system creating the box is neglected. The assumption that the particle is a closed system is not fulfilled. And again, the system does not have zero entropy.

*In short*, entropy values smaller than the system entropy limit can occur easily in calculations, especially when closure is assumed without careful checking. Such calculations do not invalidate the minimum system entropy kln2. However, there are more apparent counter-arguments.

## 10. The Minimum Entropy vs. the Extensivity of Entropy

A minimum entropy value can also appear paradoxical because a minimum value seems to contradict the *extensivity* of entropy. In everyday life, the entropy of a kilogram of water is twice the entropy of half a kilogram. Now, the minimum entropy also applies to a single atom. Therefore, an everyday system, composed of many atoms, should have an entropy value given by the minimum entropy multiplied by the number of atoms.

However, the experiments discussed above show that this is *not* the case: the entropy per particle can be much lower than kln2. In fact, this observation shows that entropy is *not extensive* in the general case.

This paradox was already known to Gibbs, as Jaynes explains [115]. Jaynes says that Gibbs understood that ‘when two systems interact, only the entropy of the whole is meaningful. Today we would say the interaction induces correlations in their states which makes the entropy of the whole less than the sum of entropies of the parts’.

The work by Tsallis [107,116] makes the same point. In his papers on the non-extensivity of entropy, Tsallis shows that the extensivity of entropy requires certain conditions on the states of the subsystems: the subsystems must not be correlated. These conditions are not fulfilled when single particles are composed to form a solid. In other words, entropy is extensive only when the subsystems are separable. A common case for which this is valid is the thermodynamic limit.

*In short*, the extensivity of entropy is only valid for non-correlated sub-systems. Therefore, the entropy per particle *can be smaller* than the smallest system entropy. The extensivity of entropy does not contradict the existence of a lowest total system entropy kln2.

## 11. What Is the Minimum Entropy Value?

At a temperature *T*, the energy *E* per degree of freedom has the typical value E=kT/2. The value k/2, which is smaller than kln2, also appears in certain uncertainty relations, as shown below. Could the smaller value be the correct quantum of entropy?

The ratio E/T is an entropy. Using the expression S=klnΩ, the entropy value k/2 corresponds to $\Omega =\sqrt{e}$, where e=2.71828⋯ is Euler’s constant. The result confirms that the entropy value k/2 is not a *system entropy*, which would require an integer number of states Ω, but an *entropy change* or (a part of) an *entropy per particle*. However, as argued above, these observables are not bounded from below. Nevertheless, k/2 is an entropy *change* that occurs frequently in physical systems.

We note that a *falsification* of the minimum system entropy is straightforward: it is sufficient to measure a smaller value than kln2 for system entropy. However, given the tight relation between *k* and the particle structure of matter and radiation, it is unlikely that this will ever happen. So far, all experiments confirmed the quantum of entropy.

In fact, an observable entropy value below kln2 has been predicted in Majorana zero modes. An example is reference [117]. However, such values below the entropy limit do not appear for (closed) systems, but only for interacting and open ‘systems’.

*In short*, it appears that there is no experimental or theoretical argument for a smallest or minimum (system) entropy smaller than kln2. We can now explore the next question posed at the beginning.

## 12. Is Total System Entropy Quantized?

When Planck explored black body radiation, he discovered, introduced, and named both the quantum of action *ℏ* and the Boltzmann constant *k*. Continuing our exploration, we can ask whether *total system entropy* is *quantized*, i.e., whether its value, even when macroscopic, is an *integer multiple* of a quantum of entropy.

The idea of the quantization of total entropy is suggested by analogy with thermal energy. Thermal energy can be considered as a multiple of kT/2. However, for total entropy, the expression S=klnΩ implies the *lack* of entropy steps. Also experimentally, entropy steps of the order of *k* in macroscopic systems have not been detected. Indeed, neither theoretical nor experimental claims about the issue are found in the literature.

The closest claim to quantization has been made for materials with a small amount of disorder at low temperatures. In this case, observable entropy steps have been predicted [118]. However, so far, no experiment confirmed the prediction.

The idea of the quantization of total entropy can also arise from an analogy with black holes, where total entropy is indeed quantized, as a result of the quantization of the area in multiples of the Planck area. However, no such argument arises for three-dimensional systems at everyday scales in flat space.

*In short*, the total entropy of three-dimensional systems composed of a macroscopic number of particles is *not* quantized in a practical sense. In experiments, the smallest entropy value only arises in systems consisting of one particle. Macroscopic systems in everyday life have effectively continuous entropy values. However, one type of system behaves differently.

## 13. Black Hole Horizons

In the domain of *quantum gravity*, the entropy of gravitational horizons—as they arise in black holes—is quantized. Many scholars have explored the quantization of black hole entropy, usually starting from the Bekenstein-Hawking entropy
(2)$$S=k\frac{A}{4G\hbar /c^{3}}\;\;.$$Many authors have argued that in black holes, entropy is quantized in multiples of a smallest value, in the same way that the area of horizons *A* is quantized in multiples of the Planck area $G\hbar /c^{3}$. One reason for the quantization of black hole entropy is that black holes, in contrast to everyday thermodynamic systems, are effectively *two*-dimensional.

The value of the quantum of entropy for black hole horizons remains a matter of debate. This value has been argued to be kln2, as was concluded in 1975 by DeWitt [103], then by Mukhanov [119], and by García-Bellido [120]. As mentioned, DeWitt also argued that kln2 is the *maximum* entropy that an elementary particle can carry, because the *least* information one can have about it is whether it exists or not, which is 1 bit. Feng et al. [121] came to the same conclusion by referencing Bekenstein [122].

In contrast, Hod [123,124] argued for an entropy quantum kln3, and explained that Bekenstein also favored this value. Instead, Kothawala et al. [125], Skákala [126,127], Maggiore [128], Liu et al. [129], Ren et al. [130], Yu and Qi [131] and Bakshi et al. [132] argued for a horizon entropy quantum of 2πk. Corishi et al. [133,134] proposed $2\gamma_{0}\;k\;\ln3$, where the Barbero-Immirzi parameter $\gamma_{0}$ is unspecified, Sakalli et al. [135] and Rahman [136,137,138] deduced more complex expressions. Liao and Shou-Yong [139] deduced 2πk/3, and Jiang [140] and Aldrovandi and Pereira [141] deduced the value *k*. The list is not exhaustive but gives an impression of the situation.

A different approach was used by Mirza et al. [142], who showed that in black holes, the emission of entropy is limited by a value of the order of *k* divided by the Planck time. Given that the Planck time is the shortest time that can be measured or observed in nature, the entropy emission limit again implies the existence of a quantum of entropy of the order of the Boltzmann constant *k*. However, no precise numerical factor has been deduced from the entropy emission limit.

The numerical prefactor in the entropy quantum in all these papers varies because, owing to the impossibility of measuring black hole entropy in experiments, a choice must be made: the number of microstates per area must be clarified. In popular accounts, the horizon area is assumed to store one bit per Planck area $G\hbar /c^{3}$; however, this choice does not agree with the expression by Bekenstein and Hawking. The situation simplifies drastically if one assumes an average of e=2.718⋯ microstates for each area $4G\hbar /c^{3}$. In this case, the quantum of entropy for black holes is simply *k*. The number of microstates per horizon area can only be settled with a theory of quantum gravity. (One such approach is presented in reference [143]).

In quantum gravity, also curved space far away from black holes is known to contain entropy and flows of entropy [144,145]. In contrast, infinite, flat, and empty space does *not* contain entropy. However, exploring the entropy of curved space yields the same issues and results as exploring the entropy of black holes: the entropy of curved space cannot be measured experimentally and the calculations yield the same discussions as those for black hole horizons.

*In short*, quantum gravity research finds that the full entropy of black hole horizons is quantized in integer multiples of O(1)k, which is also the smallest possible gravitational entropy. Horizons, which are essentially two-dimensional structures, *differ* from three-dimensional systems such as materials or photon gases, where macroscopic system entropy is effectively continuous and not quantized. It must be stressed that there are no experiments on the entropy of black holes. There is no way to experimentally check whether black hole entropy is quantized and, if so, what the exact value of the entropy quantum is. Even in so-called *analog* black holes, such as acoustic black holes or superfluid ${}^{4}$He analogs, to our knowledge, quantized entropy has not yet been measured, even though such an effect has been predicted [146,147]. Likewise, to the best of our knowledge, no discussion on the achievable measurement precision of black hole entropy has been published.

## 14. Against a ‘Quantum of Entropy’

In old Italian, the term ‘quanto’ denotes a small amount. Galileo introduced the term into physics. Planck then took over the term, and quantum physics was named. In fact, several arguments can be made against the use of the expression ‘quantum of entropy’. First, this expression is used only rarely, in a few texts on thermodynamics [1,2,3,4,5,6].

Secondly, the concept of a quantum of entropy is confusing. A quantum is usually considered as the *smallest* possible value. However, in the case of entropy, a smallest value only exists for system entropy, but not for entropy steps or entropy changes, which can be much smaller.

Third, system entropy is *not* quantized in multiples of *k* in any practical system—except possibly for black holes. Speaking of a quantum without quantization generates uneasiness. On the other hand, energy levels in atoms have quantized energy values, and these values can be extremely close. And like the possible energy levels in quantum systems, the possible entropy values in thermal systems also depend on the system details. In practice, entropy quanta are not countable in most cases—except for the cases of quantized conduction. The mentioned criticisms are also made by Blöss in his work [148].

Thus, one might prefer the expression ‘minimum system entropy’ to that of ‘quantum of entropy’. Indeed, one could consider the expression ‘quantum of entropy’ an example of modern *hype*. In this text, both expressions are used.

*In short*, the term ‘quantum of entropy’ is unusual, but no hard argument appears to exist against the use of the term. If one prefers, one can use the expressions ‘minimum system entropy’ or ‘lower system entropy limit’ instead. In physics, an expression such as ‘quantum of entropy’ is loaded with many associations. Its use only makes sense in the case that it also expresses a deeper, underlying *principle* of nature. In many domains of physics and chemistry, descriptions of natural processes using limit principles have been fruitful [149]. Therefore, in the remaining sections, we check in detail whether the quantum of entropy is an actual *principle* of thermodynamics, i.e., whether the quantum of entropy can be used to *derive* thermodynamics.

## 15. Zimmermann’s Principle of the Entropy Limit

Starting in the year 2000, Zimmermann explored the concept of the quantum of entropy in a series of five papers entitled ‘Particle Entropies and Entropy Quanta’ [1,2,3,4,5]. The series builds on his earlier work [150,151,152]. In the first paper, Zimmermann explained that one can describe, in a many-particle system, each particle as the carrier of a part of the entropy of the system. In the second paper, Zimmermann derived all the properties of the photon gas from the assumption of a quantum of entropy. In the third paper, he derived the properties of the van der Waals gas from the concept of single-particle entropy. In the last two papers, Zimmermann explored the ideal gas and the indeterminacy relation between entropy production and time.

*In short*, Zimmermann argued that statistical thermodynamics—in particular for ideal gases, real gases, and photon gases—can be deduced from the expression
(3)ΔS=O(1)k
for single particles. In all the cases he studied, the numerical factor is greater than 1. Zimmermann thus argued that the Boltzmann constant goes beyond a conversion factor between temperature and energy. All of Zimmermann’s work suggests that there is a *principle of the entropy limit*.

## 16. Thermodynamics from the Quantum of Entropy

Thermodynamics, as traditionally taught, is based on a few fundamental ideas: the existence of thermodynamic state variables, as well as the zeroth, first, second, and third law [29,31,153,154,155,156,157]. Boyling makes this point particularly clear [158]. The foundations of statistical physics confirm this structure.

Statistical physics can be seen as based on the principle of least action, on quantum theory, and on the properties of entropy. The point is clearly made by Landau & Lifshitz [159] and by Kubo [160]. Simply stated, the principle of least action implies, using Noether’s theorem, energy conservation. The quantum of action implies the particle structure of matter and radiation, and thus implies, together with their dynamics, the existence of temperature and other state variables. Thus, the zeroth and first laws of thermodynamics are consequences of the principle of least action and of the quantum of action.

The existence of a quantum of entropy expresses the particle nature of matter and radiation, and the relation between energy, entropy, and temperature. This relation is part of the first law [161]; it is also part of the zeroth law, i.e., of the existence and definition of temperature.

The second and third laws of thermodynamics concern the state variable entropy directly. The concept of entropy is best defined and thought of as disordered energy [155] or as the mixing of states [162]. The quantum of entropy includes and implies the definition of entropy. Simultaneously, the quantum of entropy includes the particle structure of matter and radiation. Using the arguments summarized in references [155,162], the quantum of entropy implies the second law.

The third law of thermodynamics states and implies that at low temperatures, most degrees of freedom of a condensed matter system are frozen. Thus, the third law follows from quantum theory [50,51,52]. In the third law, the quantum of entropy plays an indirect role, defining the measurement unit of entropy.

The arguments confirm that entropy is a fundamental quantity. In particular, entropy is more fundamental and more intuitive than the concept of heat [163,164,165]. Entropy is so fundamental that it would merit its own unit of measurement [166].

*In short*, even though the topic is not treated exhaustively here, it appears that the minimum system entropy kln2 or the quantum of entropy *k* is at the basis of all four laws of thermodynamics. When the state variables, least action, and the quantum of action are included at the foundations, all of thermodynamics is recovered [143]. The entropy limit is indeed a fundamental *principle* of thermodynamics. In fact, there are additional reasons to speak about the *principle* of the entropy limit.

## 17. Indeterminacy Relations

Statistical physics is closely related to quantum theory. The relation became clear already in the early twentieth century, when the first indeterminacy relations for thermodynamic quantities were deduced. For example, Bohr showed that temperature *T* and energy *U* obey
(4)Δ(1/T)ΔU⩾k/2.This indeterminacy relation was discussed in detail by Heisenberg and other scholars [167,168,169].

In 1992, de Sabbata and Sivaram [170] deduced the indeterminacy relation
(5)ΔTΔt⩾ℏ/k.This relation was tested and found to agree with experiments by Gillies and Allison [171,172]. In 2004, Kovtun, Son, and Starinets showed that the ratio between shear viscosity η and entropy volume density *s* follow [173,174]
(6)$$\frac{\eta }{s}⩾\hbar /4\pi k\;\;.$$In 2011, Zimmermann showed [175] that in quantum thermodynamics, entropy production *P* and time *t* obey
(7)ΔPΔt⩾k/2.A similar relation was deduced by Falasco and Esposito [176] and verified experimentally by Yan et al. [177]. Many additional indeterminacy relations exist; a comprehensive list is given by Hohm [174]. All these indeterminacy relations suggest that entropy resembles action, with a multiple O(1)k of the Boltzmann constant playing a role similar to that of *ℏ*. For example, Parker et al. [178,179] use similar versions of the entropic uncertainty relations and the quantum of entropy to calculate the configurations of alpha particles and details of cosmological systems.

Maslov [180] has extended the analogy between quantum theory and thermodynamics by defining quantum operators for internal energy, for free energy, and for entropy. In analogy to quantum theory, the measured values of these quantities are the eigenvalues of these operators.

*In short*, the quantum of entropy plays a similar role in thermodynamics as the quantum of action in quantum theory. In both cases, the minimum measurable value arises, with a factor O(1), also in indeterminacy relations. This property again underlines that *k* is not only a conversion factor but that it has a fundamental significance in thermodynamics.

## 18. Entropy Production

The explorations of entropy production in non-equilibrium systems has become an important field of enquiry. Since Jarzynski’s work [181,182] it has become possible to deduce upper limits for entropy production and to verify them experimentally [176,177,183,184,185]. In the context of this study, entropy production is a form of entropy change. In macroscopic systems, the entropy change is not bounded from below and can be as small as desired. The uncertainty relation (Equation 7) does not limit it, if the time involved is made sufficiently large. Also, experiments by Koski et al. in open systems measured extremely small entropy changes [186]. In one-particle closed systems, no exceptions to the smallest system entropy are found.

Likewise, the quantum of entropy does not contradict the limit on entropy change of quantum gravity, which, as mentioned above, is of the order of the quantum of entropy *k* divided by the Planck time [142].

*In short*, entropy production limits do not contradict the quantum of entropy *k*. In fact, the entropy production limits underline its importance: the quantum of entropy appears in all the relevant inequalities.

## 19. Similarities and Differences between Action and Entropy

The similarities between action and entropy are striking. In nature, there exists a quantum of action *ℏ* and a quantum of entropy *k*. Quantum theory, including aspects such as indeterminacy relations and entanglement, is based on the quantum of action. All the effects of quantum theory depend on the quantum of action. Thermodynamics, including the second law, is based on the quantum of entropy. All the effects of thermodynamics depend on the quantum of entropy.

The quantum of entropy plays a role in statistical thermodynamics because the quantum of entropy determines the average energy per degree of freedom. The quantum of entropy is used to define a measurement unit for temperature in the SI system. The quantum of entropy enters in all calculations of the thermodynamic properties of materials. This confirms that the entropy limit is a fundamental *principle* of thermodynamics.

Both action and entropy are extensive quantities. Furthermore, both the quantum of action and the quantum of entropy are related to the discrete structure of physical systems. Both the quantum of action and the quantum of entropy distinguish classical physics from quantum physics. There is a well-known continuum limit of thermodynamics—generally consistent with the thermodynamic limit—in which k→0 [187]. It leads to classical thermodynamics. Like the limit ℏ→0, the limit k→0 also prevents the calculation of any specific material property. All material properties are due to the quantum of action and to the quantum of entropy. The quantum of action and the quantum of entropy make similar statements: if either quantum did not exist, particles, thermal effects, and quantum effects would not exist.

The *differences* between action and entropy are also important [188]. Action and entropy resemble each other because they are both due to microscopic processes, but they differ in their relation to *change* in a system. Action describes change occurring in nature as a product of energy and time; a large amount of action implies a large *amount* of change. And despite the existence of a quantum of action, nature minimizes action in any process in an isolated system. In contrast, entropy describes the distribution of energy and change, in particular the distribution between macroscopic and microscopic change. A large amount of entropy implies a large amount of microscopic disorder. Entropy is the *cost* of change. And despite the existence of a quantum of entropy, nature maximizes entropy in processes in any isolated system.

Parker, Jeynes et al. explored the similarities and differences between action quanta and entropy quanta in detail. They also compared in detail action per time, i.e., energy, and entropy per time, or entropy production [178,179]. An important difference is that the quantum of action implies that action and action change are quantized, as observed [189,190,191,192,193,194,195,196,197,198,199,200,201,202,203,204,205,206]. In contrast, this is *not* the case for entropy.

*In short*, the quantum of action *ℏ implies* particles and describes their motion; the quantum of entropy *k results* from particles and describes their statistics. To include black holes, it can be said:

▷ The Boltzmann constant *k* expresses that everything that moves is made of discrete constituents.

In other words, the minimum system entropy, sometimes called the quantum of entropy, is a fundamental property of nature. The minimum system entropy is a limit of nature like the speed limit, the action limit, and the other limits of nature [143].

## 20. Conclusions: A Consistent Presentation of the Quantum of Entropy

The present study explored the existence of entropy quanta and of a minimum system entropy. In accordance with the third law of thermodynamics, it was shown that in thermodynamic systems consisting of a large number of particles, there is *no* smallest entropy value *per particle* and *no* smallest entropy *change*. However, there *is* a *minimum system entropy* value, which can also be called a *quantum of entropy*, that is based on the Boltzmann constant *k*:

▷ The minimum entropy limit S⩾kln2 holds for every closed physical system.

The statement agrees with all experiments. Equality is only achieved for systems consisting of a single particle. The lower limit on entropy is only in *apparent* contrast to the usual formulation of the third law of thermodynamics, to the often inappropriately claimed extensivity of entropy, and to all other apparent counter-arguments.

We also showed that any *observable* physical system must have at least *two* states. Therefore the minimum system entropy kln2 holds for *every* physical system.

Both for condensed matter and for light beams, experiments, and theory confirm that total entropy values are *not* integer multiples of the Boltzmann constant *k* or of a similar value. In contrast, in black holes, entropy *is* expected to be quantized in integer multiples of O(1)k. The reason is that the underlying constituents of black hole horizons are discrete and that in two dimensions, larger numbers of these constituents do not result in smaller entropy steps—in contrast to everyday, three-dimensional systems. In simple words, the *quantum* of entropy holds for single particles, whereas countable *quanta* of entropy only arise in black holes.

Finally, all the laws of thermodynamics can be deduced from the state variables and the minimum system entropy. The quantum of entropy also explains the indeterminacy relations between thermodynamic variables. Thus, the Boltzmann constant *k* is more than a simple conversion factor: it is a fundamental property of nature expressing that everything that moves is composed of discrete components.

In conclusion, in the same way that the speed limit *c* is a principle of special relativity and the quantum of action *ℏ* is a principle of quantum theory, the system entropy limit kln2 is also a principle of thermodynamics.

## Data Availability

No additional data is available for this work.

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
