# Peer review of "Testing the Minimum System Entropy and the Quantum of Entropy"

_entropy, 2023, doi:10.3390/e25111511_

Round 1

Reviewer 1 Report

Comments and Suggestions for Authors

please see the pdf file

Reviewer 2 Report

Comments and Suggestions for Authors

The authors argue that the entropy of a system is lower-bounded by an universal value k ln 2, which they call as entropy quantum, thus invalidating Planck’s formulation of a third law of thermodynamics. They discuss supporting arguments for such a thesis, as well as possible counterarguments. This a provocative and controversial statement, which in my opinion is not fully supported by the presented argumentation, which is usually very heuristic. However, such provocative theses may catalyze development of science, even when not accepted as a consensus of a scientific community. However, certain points must be clarified by the author before I can recommend the publication.

First, on the conceptual level, the author admit that the term “entropy quantization” may be misleading, as it does not refer to quantization of entropy changes, but to a minimum bound on an absolute value of entropy. However, this may seem to be superfluous and of no practical relevance, as in standard formulation of equilibrium thermodynamics entropy is defined only up to an additive constant, and only the entropy changes (not its absolute value) are relevant. The author do not present any practical consequences of the existence of such limit.

Second, the authors seem to misinterpret the literature to support their conclusions. For example, they argue that the idea of Landauer erasure supports their idea of a minimum bound on a system entropy k ln 2. However, in its standard formulation, the Landauer erasure corresponds to transforming an indeterminate state of a bit with a maximum (!) entropy k ln 2 to a definite state with zero entropy. This requires the heat dissipation of at least Q = k_B T \ln 2, such that the entropy production \Delta S+QT is nonnegative. To quote an original paper of Landauer “The

well-defined initial state corresponds, by the usual statistical mechanical definition of entropy, S=k log, W, to zero entropy”. The author must thus justify, why in their opinion concept of Landauer erasure supports their thesis.

The authors argue also that the heat conductance quantization supports the idea of entropy quantization. However, they do not justify this statement adequately. For example, thay state that “Meschke et al. observed that heat transport via a sequence of single photons confirms the quantization of heat conductance. [55] They found that each photon carries an entropy of the order k”. However, the conductance quantization is not a generic occurrence even in quantum transport, but rather corresponds only to the case of ballistic transport through channels with a perfect transmission. This corresponds to the case of a matched circuit from Ref. [55]. As implied by Eq. (3) from [55], in a generic case of an unmatched circuit, the heat conductance is not quantized, i.e., it can be an arbitrary noninteger multiple of a conductance quantum. Furthermore, the sentence “They found that each photon carries an entropy of the order k” is not supported by any direct statement from Ref. [55], nor by any reasoning providing by the author.

Finally, the argument of the authors is clearly at odds with a modern formulation of thermodynamics of small systems within the framework of stochastic thermodynamics. Applicability of this approach has been experimentally well confirmed, i.e., by experimental tests of a fluctuation theorem for entropy production in J. V. Koski et al., Nat. Phys. 9, 644 (2013). In this formulation entropy of a system is identified with a Shannon entropy, which corresponds to a Boltzmann entropy at equilibrium. For a two-state system this entropy can take values between 0 and k \ln 2, and is thus actually upper- and not lower-bounded by the quantum of entropy argued by the authors.

Round 2

Reviewer 1 Report

Comments and Suggestions for Authors

The authors have made substantial improvement on the manuscript, following the comments of mine and another reviewer. I consider the present version of the manuscript is qualified for publication.